# ANALYTICAL SOLUTIONS FOR A FAMILY OF SINGLE LAYER NEURAL NETWORK REGRESSION PROBLEMS

**Siddharth Krishna Kumar**
Upwork Inc.
siddharthkumar@upwork.com

## ABSTRACT

In this paper, we analyze a family of penalized single layer neural network regression problems wherein the response variable has all non-negative entries. We show analytically that the optimal weights of the problem lie at the vector of zeros, which is a point of non-differentiability.

## INTRODUCTION

Theoretical advancements in the field of neural networks Ahn et al. (2022); Arora et al. (2022); Reddi et al. (2019); Ziyin et al. (2021) have made substantial contributions to our comprehension of these models. However, most of these studies have focused on neural networks that are continuously differentiable, while neural networks often incorporate non-differentiable components. In this paper, we extend the existing literature by constructing a family of neural network regression problems where the optimal solution aligns with the vector of zeros – a point of non-differentiability. By examining these problems, we aim to further explore the impact of non-differentiability within neural networks.

## PROBLEM SETUP

Consider the penalized single layer neural network which regresses a vector $\mathbf{y}$, on a data matrix $\mathbf{X}$, using a single layer neural network with RELU non-linearities is given by

$$f(\beta) = ||y - \max(0, \mathbf{X}\beta)||_2^2 + \lambda_1||\beta||_1 + \lambda_2||\beta||_2^2, \tag{1}$$

where $\mathbf{X}$ is an arbitrary data matrix, $\mathbf{y}$ is a non-positive vector (i.e., $\mathbf{y}[i] = -\alpha_i$ with $\alpha_i \geq 0$ for all i), $\lambda_1 \geq 0$ and $\lambda_2 \geq 0$. In this paper, we show analytically that when at least one of $\lambda_1$ or $\lambda_2$ is strictly greater than 0, then $\beta^* = [0, 0, \ldots, 0, 0]^T$ is the unique minimizer of (1)

## PROOF

With the $i^{th}$ row of $\mathbf{X}$ given by $\mathbf{X}_{i,:} = x_i^T$, and $\mathbf{y}[i] = -\alpha_i$ with $\alpha_i \geq 0$ as described above, define

$$h_i(\beta) = (\alpha_i + \max(0, x_i^T\beta))^2 \tag{2}$$

and

$$g(\beta) = \lambda_1||\beta||_1 + \lambda_2||\beta||_2^2. \tag{3}$$

With a minor re-arrangement of the terms, equation (1) can be rewritten as

$$f(\beta) = \sum_{i=1}^{i=N} h_i(\beta) + g(\beta) \tag{4}$$

Our proof proceeds in two parts. First, we show that $f(\beta)$ is a convex function of $\beta$, and therefore, every local optimal solution is also a global optimal solution. Next, we show that

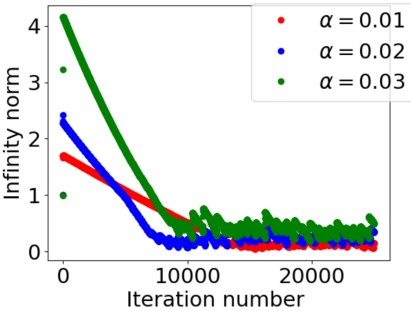

Figure 1: The infinity norm of the gradient descent trajectory for the loss function described in (1). The entries in the data matrix are randomly sampled from $[-1, 1]$, and the entries in the response vector are sampled from $[-1, 0]$

$\beta^* = [0, 0, \ldots, 0, 0]^T$ minimizes $f(\beta)$, and that for every $\beta' \neq \beta^*, f(\beta') > f(\beta^*)$. As a result, $\beta^*$ is the unique minimizer of $f(\beta)$. We omit proofs for the well established facts that $g(\beta)$ is a convex function of $\beta$, and that $\beta^* = [0, 0, \ldots, 0, 0]^T$ is the unique minimizer of $g(\beta)$ whenever $\lambda_1 > 0$ and/or $\lambda_2 > 0$.

**Proposition 1** *$f(\beta)$ is a convex function of $\beta$*

**Proof** Using example 3.5 (page 80) and section 3.2.1 (page 79) in Boyd & Vandenberghe (2004), we have that $(\alpha_i + \max(0, x_i^T \beta))$ is a convex function of $\beta$. Furthermore, since $(\alpha_i + \max(0, x_i^T \beta)) \geq 0$ for all values of $\beta$, (2) implies $h_i(\beta)$ is a convex function of $\beta$ for every $i$ (bullet point 4 in example 3.13 (page 86) of Boyd & Vandenberghe (2004)). Finally, since $g(\beta)$ is a convex function of $\beta$, (4) implies that $f(\beta)$ is a convex function of $\beta$. Therefore, any locally optimal point in $f(\beta)$ is also globally optimal
∎

**Proposition 2** *$\beta^* = [0, 0, \ldots, 0, 0]^T$ is the unique global minimizer of $f(\beta)$*

**Proof** From (2), $h_i(\beta)$ attains its minimum value whenever $\max(0, x_i^T \beta) = 0$. Since $\beta^*$ satisfies this condition, $\beta^*$ minimizes $h_i(\beta)$ for every $i$. Furthermore, since $\beta^*$ also minimizes $g(\beta)$, (4) implies $\beta^* = \arg\min_\beta f(\beta)$.

To prove the uniqueness of the solution, we note that when $\lambda_1 > 0$ and/or $\lambda_2 > 0$, $\beta^*$ is the unique minimizer of $g(\beta)$. Therefore for any $\beta' \neq \beta^*$, we have $g(\beta') > g(\beta^*)$ and $h_i(\beta') \geq h_i(\beta^*)$ for every $i$. Combining the inequalities, we have $\sum_{i=1}^{i=N} h_i(\beta') + g(\beta') > \sum_{i=1}^{i=N} h_i(\beta^*) + g(\beta^*)$, i.e., $f(\beta') > f(\beta^*)$. ∎

## EXPERIMENTS

To validate our claim, we first generate an arbitrary $\mathbf{X}$ of dimension $20 \times 500$ whose entries are uniformly sampled in $[-1, 1]$. Next, we generate a $20 \times 1$ vector $\mathbf{y}$ with entries uniformly sampled on $[-1, 0]$. Finally, we initialize weights uniformly on $[-1, 1]$, and run gradient descent on (1) with $\lambda_1 = 0.01$ and $\lambda_2 = 0.001$ for different values of $\alpha$. We observe that in every scenario, the infinity norm of the solution consistently remains close to 0, aligning with our expectations.

## URM STATEMENT

The authors acknowledge that at least one key author of this work meets the URM criteria of ICLR 2023 Tiny Papers Track.

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
