# OpenReview forum: "Analytical solutions for a family of single layer neural network regression problems "
_ICLR.cc/2023/TinyPapers — Submitted to Tiny Papers @ ICLR 2023_

### Official Review · Reviewer_CGhy · 2023-03-26

**Confidence:** 3

**Summary Of Contributions:**

Optimal weights for regression problems using single layer neural networks were shown as [0, 0, . . . , 0, 0]T  under certain conditions. Relevant proofs were shown in this work.

**Rating:**

Clear, Correct, and Reproducible (CCR): a submission which meets the reviewing criteria

**Strengths And Weaknesses:**

Strengths :

*  The derived proofs were crisp, simple and were built on top of existing work
*  Relevant work from standard text book was cited

Weaknesses

*  Additional empirical evaluations to support the proofs can always enhance this work
*  This work covers a specific case, however any opportunities to generalise this to other settings can enhance the impact

**Suggested Changes:**

n/a

---

> ### Author Response · Authors · 2023-05-24
> **Response to CGhy**
>
> Thank you for your kind reviews. We have added simulations for emperical support of our results. We are currently working on generalizations of the current result, but we could not do them justice in the two page limit; they will be part of a longer followup paper

---

### Official Review · Reviewer_3kTw · 2023-03-29

**Confidence:** 2

**Summary Of Contributions:**

The authors provide an analytical solution for a family of single layer NNs applied to regression problems. Their solution is valid when y is non-positive and when some other constraints are met.

**Rating:**

Needs Clarification (NC): a submission which does not meet the reviewing criteria and needs clarification for its described problem or solution

**Strengths And Weaknesses:**

I'm not particularly familiarized with convex optimization. That's a hard constraint on the quality of my review, so I won't be able to entirely validate your proof or talk about the math of it all. I apologize for that. That being said, I'd like you to take my review as it is: from someone who knows close to nothing of this area and expected to get some basic intuitions from reading the paper.

I believe the main weakness of this tiny paper is that it gives readers no context on why this particular family of NNs and the proof itself are important for the machine learning community. It'd be nice to have some preliminaries for a total beginner in the subject as me, so that we can understand, at least partially, the importance and strength of this submission.

**Suggested Changes:**

- Please add some preliminaries and expected impacts this analytical solution might have to the community if possible. As I mentioned in the strengths/weaknesses section, as I'm not experienced in this topic, the amount of knowledge I got from the proof alone was extremely scarce.
- Are there any "toy problems" or some more intuitive application you could add to exemplify its use cases and importance?

---

> ### Author Response · Authors · 2023-05-24
> **Response to 3kTw**
>
> Thank you for your kind reviews. We have rewritten our abstract and introduction, with references detailing why we feel this problem is important.

---

### Author Response · Authors · 2023-05-30
**Opt in for archival**

I wish to opt-in for archival of this paper

---

### Meta-Review · Area_Chair_JWah · 2023-04-05

**Recommendation:** Invite to archive
**Confidence:** 4

**Metareview:**

Pros:
- All proofs are included, making the work reproducible.
- The results are correct.
- An interesting (simple) observation about optimization of single layer ReLU networks is obtained.

Cons:
- The derived result has not been placed in context of any related known results, which affects clarity of writing.
- What is the role/significance/implication of y having all non-positive entries?
- The result essentially point out that the global optimum for the single layer model/optimization-problem considered is not very interesting; and this issue only occurs when all labels are strictly negative, which is rare and easily avoidable.

**Summary:**

The unique optimal weights of a single-layer neural network regression (with L1 and L2 regularization) problem are derived. Correct proofs but missing relevant literature.

**Comments And Feedback To The Authors:**

It would be good to include:
- More context on why the problem is important, and connection to known comparable results (e.g. a similar result also holds for the elastic net regression with squared loss).
- As another reviewer points out, it could be interesting to prove extensions of the current result, say along the lines of studying other losses (instead of squared loss) or activations.

Additional minor writing suggestions:
- Consider reproducing the examples from the Boyd and Vandenberghe book (say, in an appendix) for reader's convenience.
- Boyd repeated in citation text "Stephen Boyd, Stephen P Boyd, and Lieven Vandenberghe..."

**Reason For Not Giving A Higher Recommendation:**

A main limitation is lack of discussion on relevant literature.

**Reason For Not Giving A Lower Recommendation:**

The results are correct and reproducible.

---

> ### Author Response · Authors · 2023-05-24
> **Response to JWah**
>
> Thank you for your kind reviews. We have rewritten the abstract and introduction to address your concerns. We have also fixed the references as you suggest.
>
> As we mention in our rewrite, the problem is interesting because it provides analytical solutions to a family of problems where the optimal solution lies at the point of non-differentiability. This is in contrast to most theoretical results which focus on the analyses of continuously differentiable loss functions.
>
> We are currently working on generalizations of the current result, but we could not do them justice in the two page limit; they will be part of a longer followup paper

---

### Decision · Program_Chairs · 2023-04-08

Invite to archive